# Synaptic potentiation of engram cells is necessary and sufficient for context fear memory

Leonardo M. Cardozo [1,2,3,4] ✉, André F. de Sousa [2,3,5,7] ✉, Blythe C. Dillingham[2,3,5], Westley Dang[2,3,5], Nicholas Job[2,3], Eun J. Yoo[2,3], Sural K. Ranamukhaarachchi[2], Qi Yuan [2,6] ✉ & Mark Mayford[2,3]

The nature and distribution of the synaptic changes that underlie memory are not well understood. Here we examine the synaptic plasticity behind context fear conditioning in male and female mice and find that new learning produces synaptic potentiation specifically onto engram neurons in the basolateral amygdala. This potentiation lasts at least 7 days, is reversed by extinction, and its disruption impairs memory recall. High frequency optogenetic stimulation of the CS and US-activated ensembles, or biochemical induction of synaptic potentiation in US-responsive neurons alone, is sufficient to produce a context fear association without prior associative training. These results suggest that plasticity of CS inputs onto US-responsive amygdala neurons underlies memory formation and is necessary and sufficient to establish context fear associations.

Memories are thought to be encoded by learning-induced synaptic changes distributed within the circuits controlling behavior. In contextual fear conditioning (CFC), lesion and pharmacological studies support a role of many brain areas in memory encoding and retrieval, including the hippocampus, retrosplenial cortex, prefrontal cortex, and basolateral amygdala (BLA)[1–7]. In these brain regions, neurons active during learning and expressing the immediate early gene *cfos*, can be artificially reactivated to induce apparent memory recall, suggesting that they may be part of the so called engram neurons[8] that encode a CFC memory[9–12]. While these studies define the circuitry controlling behavior and suggest a widely distributed network supporting CFC memories, they do not identify the crucial sites of neuronal plasticity that are necessary for memory encoding.

Previous studies of tone-cued fear conditioning have found that synapse-specific potentiation of auditory inputs to the lateral amygdala (LA) is necessary for auditory fear memory and discrimination[13,14], although not sufficient on its own[15]. In contrast, the role of amygdala plasticity in CFC learning is less well characterized. Interestingly, in many of the aforementioned studies examining learning-induced plasticity in the LA/BLA, the behavioral training would be expected to produce a CFC memory in addition to a tone-conditioned memory. However, the observed synaptic changes were dependent on tone-shock pairing contingencies[16–19]. This could reflect either a lack of amygdala-based plasticity underlying context-

fear associations or a sparse context representation not captured by recording from randomly targeted neurons.

To address these questions, we employed multiple transgenic mouse lines along with biochemical, optogenetic, and electrohysiological techniques to explore the nature of the synaptic changes induced by a CFC memory in BLA neurons. We found that cfos-positive neurons activated during learning in the BLA undergo long-lasting synaptic potentiation (up to 7 days) at the time of learning that can be reversed by memory extinction. Such potentiation was both necessary and sufficient to maintain a long-term association between a spatial context and an aversive foot shock. Additionally, synaptic potentiation and memory formation could be recapitulated by artificially stimulating specific inputs to the BLA. Together, our results illuminate the cellular and molecular mechanisms underlying key synaptic processes that enable the formation of long-lasting contextual fear memories.

## Results

### Context fear conditioning induces specific synaptic potentiation in learning-activated neurons of the BLA

To test whether CFC induces specific synaptic changes in putative memory-encoding neurons in the BLA, we used the cfos-shEGFP transgenic mouse line (Fig. 1A), which expresses a short half-life EGFP under the control of the

[1]Neurosciences Graduate Program, University of California San Diego, La Jolla, CA, 92092, USA. [2]Department of Psychiatry, University of California San Diego, La Jolla, CA, 92092, USA. [3]Department of Molecular and Cellular Neuroscience, Dorris Neuroscience Center, The Scripps Research Institute, La Jolla, CA, 92037, USA. [4]CAPES Foundation, Ministry of Education of Brazil, Brasília, DF, 70040-020, Brazil. [5]Doctoral Program in Chemical and Biological Sciences, The Scripps Research Institute, La Jolla, CA, 92037, USA. [6]Division of Biomedical Sciences, Faculty of Medicine, Memorial University of Newfoundland, St. John's, NL, A1B 3V6, Canada. [7]Present address: Department of Neurobiology, University of California Los Angeles, Los Angeles, CA, 90095, USA. ✉e-mail: leominete@gmail.com; asousa@mednet.ucla.edu; qyuan@mun.ca

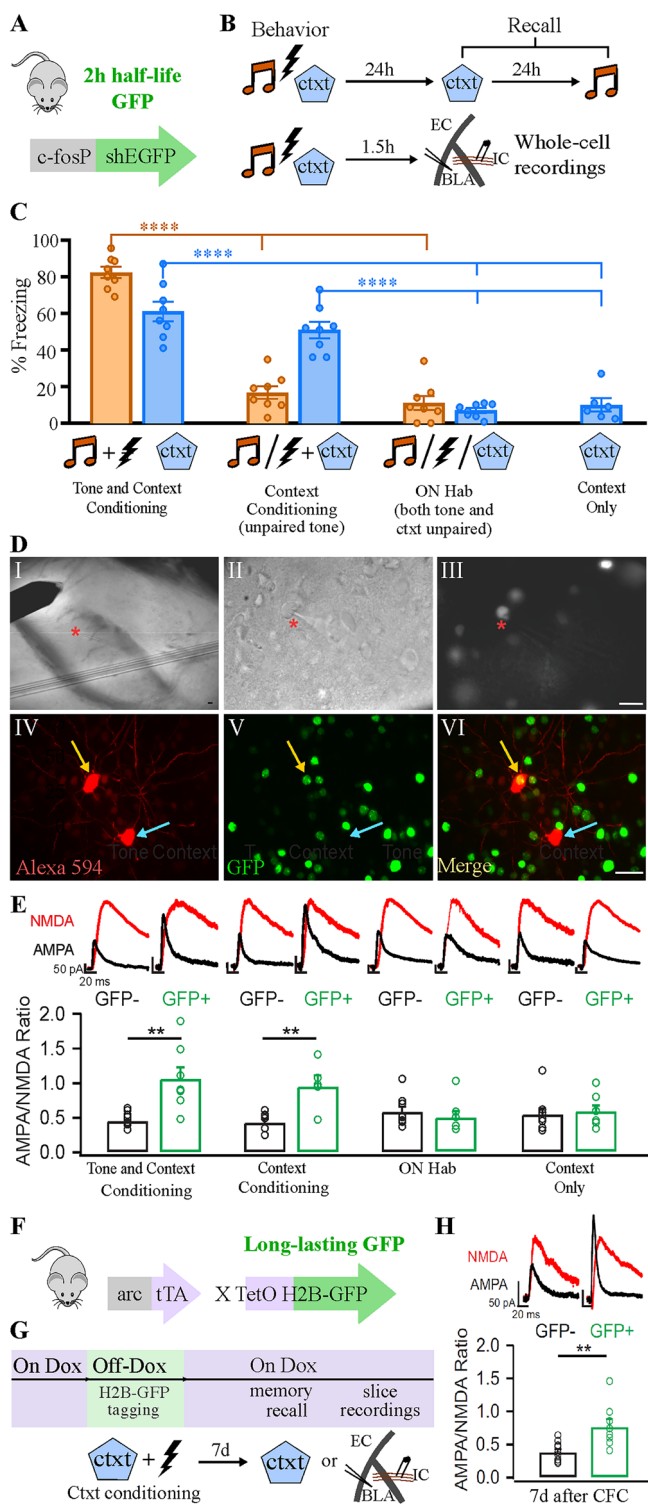

**Fig. 1 | Context learning induces long-lasting synaptic potentiation specifically in BLA neurons encoding a fear memory. A** The cfos-shEGFP transgenic mouse, expressing a short half-life (sh)EGFP under the control of the cfos promoter.
**B** Experimental design in which the cfos-shEGFP mouse was either used for whole-cell recordings or memory recall tests after different behavior treatments. Ctxt context, EC external capsule, IC internal capsule, BLA basolateral amygdala.
**C** Freezing response to the context (blue bars) and/or tone (orange bars) measured 24 h after correspondent behavior treatment. $N = 8$ per group, except in "Context only" group, in which $N = 6$. **D** Whole-cell recording design. Panel I shows stimulating electrode over internal capsule fiber bundle. II and III illustrate ongoing whole-cell recording (red asterisk) of a GFP$^+$ neuron. IV-VI are confocal images of neighboring neurons filled with Alexa 594 during recordings: the GFP$^+$ neuron is highlighted by a yellow arrow, while the GFP$^-$ is highlighted by a cyan arrow. Scale bars: 50 μm. These images were taken from a slice of an Arc-tTA/TetO-H2B-GFP mouse (**F**). **E** AMPAR/NMDAR ratio of GFP$^+$ vs GFP$^-$ BLA neurons from acute brain slices of cfos-shEGFP mice sac'ed 90 min after behavior treatment. An increase in AMPAR/NMDAR ratio was only observed after successful conditioning and was restricted to GFP$^+$ neurons. $N = 6$–9 per group. **F** Arc-tTA x TetO-H2BGFP double transgenic mouse, in which the tetracycline transactivator (tTA) was knocked in the Arc gene and controls the expression of the long-lasting histone-bound GFP.
**G** Experimental design illustrating the tagging window of the Arc-tTA/TetO-H2B-GFP mouse line, controlled by the presence of Doxycycline (Dox) in the food. Tagged neurons were recorded 7 days after. Correspondent behavior result can be found on Supplementary Fig. 2C. **H** Specific AMPAR/NMDAR ratio increase in GFP$^+$ neurons following conditioning is maintained for at least 7 days. CFC Contextual fear conditioning. $N = 8$, 9 per group. **\*\*$P < 0.01$, \*\*\*\*$P < 0.0001$, one-way ANOVA with Tukey test (**C**), unpaired t test [(**E**) and (**H**)]. Graph bars show mean $+/-$SEM.

stimulating internal capsule input fibers (Fig. 1D). GFP$^+$ cells were observed throughout the anterior to posterior BLA. We recorded α-amino-3-hydroxy-5-methyl-4-isoxazolepropionic acid receptor (AMPAR)/N-methyl-D-aspartate receptor (NMDAR) ratios as a measure of excitatory synaptic strength[17], and found that fear conditioning produced a selective increase in AMPAR/NMDAR ratio only in GFP$^+$ neurons (Fig. 1E).

Remarkably, we also observed an increase in AMPAR/NMDAR ratio in cfos$^+$ BLA neurons (GFP$^+$) when a different group of mice was trained solely in a CFC task without tone presentations, suggesting that synaptic potentiation also occurs in these BLA neurons after context conditioning (Fig. 1E). Despite this increase in the AMPAR/NMDAR ratio, GFP$^+$ neurons did not differ in spike number, latency to spike, resting-membrane potential or paired pulse ratio (Supplementary Fig. 1). Importantly, we confirmed that the observed increase in AMPAR/NMDAR ratio was driven by the association between the context (conditioned stimulus; CS) and the foot shock (unconditioned stimulus; US) since a protocol that did not induce CFC[21], involving overnight habituation to the context (ON hab; Fig. 1C), failed to induce an increase in the AMPAR/NMDAR (Fig. 1E), despite tagging a similar number of cells with GFP in the BLA (Supplementary Fig. 2A, B).

Contextual fear memories are known to form rapidly, be context specific, and persist for several days[22]. To examine the persistence of the plasticity changes observed in BLA learning-activated neurons after CFC, we employed a double transgenic Arc-tTA/TetO-H2BGFP mouse line (Fig. 1F), in which a long-lasting histone-bound GFP is expressed in Arc$^+$ cells (another immediate early gene that has been used to access memory-encoding neurons). Active neurons were tagged with GFP at the time of learning, and acute slices were prepared for whole-cell recordings 7 days later (Fig. 1G). The percentage of GFP$^+$ cells was similar to that observed in the cfos-shEGFP transgenic mouse used in previous experiments (Supplementary Fig. 2A, B). Using this approach, we found that the increase in the AMPAR/NMDAR ratio in learning-activated neurons persisted for at least 7 days (Fig. 1H). Moreover, this potentiation was reversed in a group that received extinction training (Supplementary Fig. 2C–F). Interestingly, re-conditioning animals that had undergone extinction reinstated the AMPAR/NMDAR ratio increase in the original GFP$^+$ neuronal ensemble (Supplementary Fig. 2F) suggesting that the changes in synaptic

*cfos* promoter across the entire brain[20]. Given the widely studied role of the BLA in tone-fear conditioning, we first employed a protocol that induces both tone- and context-fear associations, so we could use tone-fear conditioning as a positive control. Thus, mice underwent tone fear conditioning in a particular context and were either tested for memory 24 h later in the same context or sacrificed after 90 min to perform whole-cell recordings in a BLA slice preparation (Fig. 1B, C). Since it is unknown how contextual information reaches the amygdala, and given the likely distributive nature of such a complex stimulus, we examined synaptic responses of GFP$^+$ (cfos-expressing) and GFP$^-$ neurons with targeted recordings in the BLA while

potentiation observed in these BLA neurons are directly correlated with memory persistence.

Together, these results suggest that learning-induced synaptic potentiation in BLA neurons following CFC is circuit-specific, long-lasting, and may support memory formation.

## Learning-induced synaptic potentiation in BLA cfos+ neurons is necessary for memory recall

To test whether the synaptic potentiation described above represents a necessary component of the CFC memory trace, we used a biochemical approach to reverse these synaptic changes following learning. We employed a *cfos* driven/inducible Cre recombinase mouse line (FDC mouse)[23] combined with AAV-based gene delivery to introduce the LTD-inducing CaMKIIα-T286D (CK2-D) mutant[24] into BLA neurons activated during learning (Fig. 2A). CaMKIIα is a critical component in the induction of synaptic plasticity, with mutant forms known to either induce long term-potentiation (LTP) or long-term depression (LTD), as with the CK2-D mutant[24]. In the FDC mouse, Cre-recombinase function is regulated by trimethoprim (TMP)- induced protein stabilization, providing a ligand-gated time window of approximately 4 h for Cre activity[23,25].

Using this system, we induced expression of CK2-D and a GFP marker in BLA neurons activated during a tone- and context-fear condition task (Fig. 2B, C). We tested both tone- and context-fear memory on the same day, before significant CK2-D expression, and again 7 days later when CK2-D expression peaked (Fig. 2D; Supplementary Fig. 3A–E). As a control, another group of mice was injected with a wild type version of the CaMKIIα protein (WT-CaMKIIα) and underwent the same behavioral schedule. When tested 4 hours after learning, both groups demonstrated robust freezing to the tone and the context, indicating no immediate impairment of memory retrieval. In contrast, 7 days after learning, we observed significant impairment in both tone- and context-fear memory in mice expressing the CK2-D mutant in BLA neurons, but not in mice expressing the WT-CaMKIIα, suggesting that the observed impairment was not due to CaMKIIα overexpression (Fig. 2F, G). Interestingly, this impairment persisted when the same mice were tested again four weeks after training, suggesting that the CK2-D-induced memory impairment is long-lasting (Supplementary Fig. 4).

Staining for endogenous cfos protein during memory retrieval showed that the reactivation of neurons tagged during training (GFP+), which has been shown to correlate with memory retrieval[20,26], was impaired by CK2-D expression (Fig. 2H, I). Importantly, CK2-D expression in BLA neurons tagged during training reversed the previously observed long-lasting increase in the AMPAR/NMDAR ratio [(Fig. 2E), compare with Fig. 1H], and reduced the amplitude of mini-EPSCs without affecting their frequency, paired-pulse ratio or intrinsic excitability (Supplementary Fig. 5 and Table 1). These results suggest that the observed behavioral impairment is likely due to a specific disruption of learning-induced synaptic potentiation in BLA cfos+ neurons.

Finally, to test whether the observed CK2-D-mediated memory impairment was context-specific, we genetically tagged cfos+ neuronal ensembles activated in either the conditioning box (box A) or a novel box (box B) one day after training (Fig. 2J). When tested for recall in box A 7 days later, we found that the memory impairment occurred only when CK2-D was expressed in neurons activated in box A (Fig. 2K), despite a similar proportion of labeled neurons in both groups (Supplementary Fig. 6). Together, these results suggest that synaptic potentiation in learning-activated BLA neurons is necessary for the long-lasting maintenance of CFC memories, and that this plasticity is context specific.

## Learning-induced synaptic potentiation in BLA cfos+ neurons is sufficient for context-fear association

Next, we sought to test whether synaptic potentiation alone in learning-activated BLA neurons is sufficient to produce a context fear association. We attempted to produce a de novo associative fear memory by inducing plasticity in neurons activated independently by context exposure (CS) and

the foot shock (US), as shown in Fig. 3. We used the rAAV2-retro serotype[27] to introduce a Cre-dependent ChR2(H134R)-EYFP (ChR-retro) into the BLA of FDC mice, enabling genetic tagging of CS and US-activated ensembles in both local and in presynaptic neurons (Fig. 3A). Co-injection of an rAAV-DJ containing a Cre-dependent nuclear marker confirmed that the injection was restricted to the BLA (Supplementary Fig. 7A–C). In addition to the BLA, we found significant ChR-retro expression in (from the most intense to the least) the prefrontal cortex (PFC, including prelimbic, infralimbic and mediorbital cortices), the entorhinal cortex (EC, both lateral and medial portions), the paraventricular nucleus of the thalamus (PVT), the ectorhinal and perirhinal cortices (Ect-PRh), and the insular cortex (InC) (Fig. 3B and Supplementary Fig. 7). There was inconsistent labeling of the auditory cortex (AuC) when the injection hit more dorsal parts of BLA (Fig. 3B). Minimal to no labeling was found in some known BLA input regions such as the ventral hippocampus (vHPC)[28,29], the medial geniculate nucleus (MGN) and the periaqueductal gray (PAG), suggesting that the rAAV2-retro serotype infects cortico-amygdalar inputs more effectively (Supplementary Fig. 7E). This specificity may have derived from the directed evolution and selection process used in its development, favoring the labeling of cortical projections[27].

To test whether potentiation of either CS or US inputs, or both, is sufficient to produce a de novo associative fear memory, we used an unpaired training protocol in which the CS (box B + tone) and US (shock) were presented separately on different days to avoid any natural learned association (Fig. 3C, D). To induce synaptic potentiation in BLA inputs, we delivered 100 Hz light pulses through optic fibers implanted bilaterally over the BLA. This stimulation protocol has been shown to induce LTP in several brain regions[15,30–32]. Remarkably, the optogenetic stimulation at 100 Hz in the BLA produced a de novo associative fear memory to box B only when both CS and US inputs were tagged, while no memory was formed when either one of them was tagged alone (Fig. 3E). This artificial memory association was context-specific, as no memory to an untagged box (box C) was formed. No auditory fear association was formed, possibly due to insufficient ChR2-retro expression in auditory regions (Supplementary Fig. 8D). Indeed, when using a double transgenic line that robustly expresses the ChR2 variant ChEF in cfos+ neurons broadly throughout the brain[9], we found that the same optogenetic stimulation protocol at 100 Hz produced a de novo fear association to both the tone and context (Supplementary Fig. 9). Importantly, the optogenetic stimulation used here did not cause any noticeable performance alterations in anxiety tests, including the elevated plus-maze, the open field, and marble burying tests (Supplementary Fig. 8A–C).

One model of fear learning posits that it occurs through the potentiation of CS inputs onto amygdala neurons[33], suggesting that the potentiation of CS inputs alone would be sufficient to generate a de novo contextual fear memory association, which did not occur in our hands (Fig. 3D, E, "CS" group, in blue). One possible explanation is that the indiscriminate potentiation of CS inputs might result in no clear behavioral outcome[15], as BLA is a critical component of behaviors of distinct emotional valence[17]. Indeed, BLA neurons tagged during box exposure (CS) project less heavily to the central amygdala (CeA), a known fear center, and more heavily to the nucleus accumbens (NAc) core than BLA neurons tagged during context conditioning (CS + US) (Supplementary Fig. 10), which suggests that different neuronal populations are activated by the CS alone compared to CS + US[34]. Thus, in our setup, the tagging of US neurons might be crucial to direct the optogenetic manipulation towards aversive behaviors.

To test whether we could reverse the artificial memory association to box B, we used 1 Hz optogenetic stimulation (Fig. 3C), which has been shown to induce LTD in auditory inputs to the LA[14,15]. After applying 1 Hz light pulses, we observed that the freezing levels to box B returned to baseline, suggesting a reversal of the artificial context memory (Fig. 3F). Interestingly, the memory to box A (tagged with the US, Fig. 3C) was not significantly affected (Supplementary Fig. 8E), which suggests that natural memory recall might occur through routes that were insufficiently tagged by

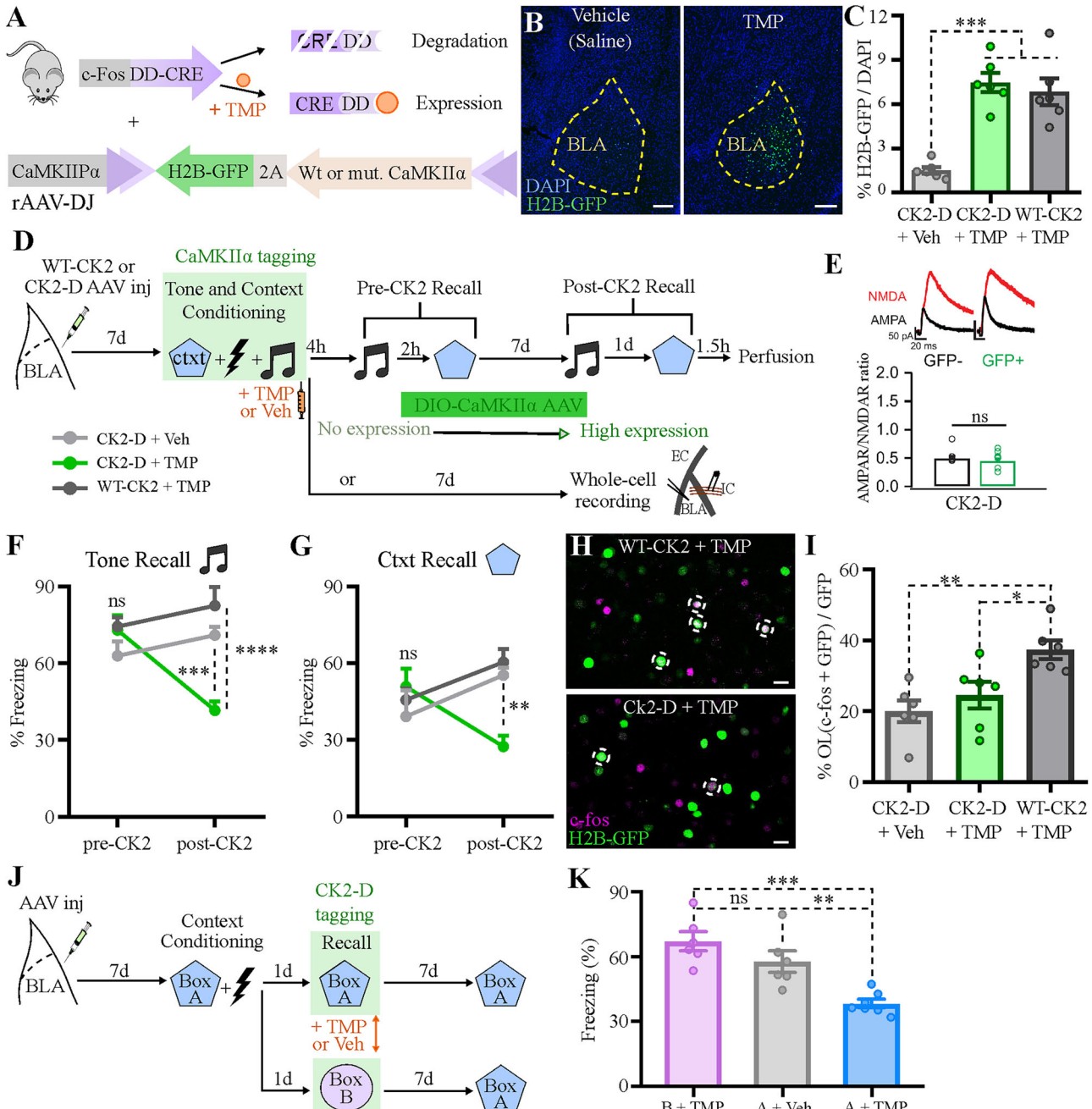

**Fig. 2 | Disruption of learning-induced synaptic potentiation in BLA engram neurons impairs long-term memory. A** rAAV-DJ carrying a DIO-(wild-type or mutant) CaMKIIα-2A-H2BGFP construct was injected bilaterally in the BLA of cfos-DD-Cre (FDC) mice, in which Cre-recombinase is fused to a destabilizing domain (DD), which leads to protein degradation. The drug trimethoprim lactate (TMP) stabilizes the complex, allowing Cre to activate the CaMKIIα construct along with the H2B-GFP marker in cfos[+] BLA neurons. **B, C** Representative coronal BLA sections showing H2B-GFP induction 14 days after TMP or vehicle (saline) IP injection, quantified in (**C**). $N = 6$ mice per group. Scale bar: 200 μm. **D** Experimental design to test whether CK2-D impairs memory recall. **E** CK2-D expression in cfos[+] BLA neurons tagged during context conditioning reverses learning-induced AMPAR/NMDAR ratio increase. No tone conditioning/exposure was performed in

order to allow for clearer interpretation of the effect of CK2-D expression on context conditioning-induced plasticity. $N = 7$, 8 neurons per group. **F, G** CK2-D expression in cfos[+] BLA neurons tagged during training impairs long-term memory, as opposed to WT-CK2. $N = 6$ mice per group. **H, I** Representative coronal BLA sections showing overlap between activity markers during training (H2B-GFP) and context recall [(cfos IHC), (**H**)], quantified in (**I**). CK2-D reversal of synaptic potentiation impaired reactivation of learning-induced neurons. OL: overlap. $N = 6$ mice per group. Scale bar: 20 μm. **J** Experimental design to test context specificity. **K** CK2-D-mediated memory impairment is specific to tagged context. $N = 6$, 7 per group. *$P < 0.05$, **$P < 0.01$, ***$P < 0.001$, ****$P < 0.0001$, ns not significant; one-way ANOVA with Tukey test [(**C**), (**I**) and (**K**)], unpaired t-test (**E**), or two-way RM ANOVA with Tukey test [(**F**) and (**G**)]. Graphs show mean $+/-$SEM.

ChR2-retro, including vHPC[29] and subcortical regions (Supplementary Fig. 7C). Despite the need for tagging both CS and US inputs for the generation of a de novo memory, we found that the tagging of the US neurons alone was sufficient to lead to light-induced freezing during prolonged optogenetic stimulation at 20 Hz in a novel box (Fig. 3G). This stimulation

protocol also induced enhanced cfos expression in the BLA, which overlapped with the US-tagged cells (Supplementary Fig. 11).

One caveat of optogenetic stimulation is the potential off-target effects both upstream and downstream of the target area due to the stimulation-driven circuit response[35]. Therefore, based on the optogenetic experiments

https://doi.org/10.1038/s42003-025-08140-6                                                                                       **Article**

**Table 1 | CaMKIIα-T286D (CK2-D) expression in cfos⁺ BLA neurons tagged during training does not affect intrinsic electrophysiological properties**

|  | CK2-D⁻ | CK2-D⁺ | t | p |
|---|---|---|---|---|
|  | **n = 16** | **n = 10** |  |  |
| AP threshold (mV) | −44.3 ±1.1 | −44.4 ±1.6 | 0.04 | 0.97 |
| AP amplitude (mV) | 74.7 ±2.8 | 72.8 ±3.7 | 0.41 | 0.69 |
| AP half width (ms) | 0.96 ±0.09 | 0.85 ±0.08 | 0.8 | 0.43 |
| V resting (mV) | −74.1 ±1.9 | −74.3 ±2.8 | 0.07 | 0.94 |
| Rm (Mohm) | 222.2 ±28.9 | 210.4 ±32.7 | 0.26 | 0.79 |
| AP numbers at 300pA | 4.5 ±0.5 | 4.1 ±0.9 | 0.43 | 0.67 |
| Latency to spike (ms) | 6.6 ±0.8 | 5.3 ±0.7 | 1.12 | 0.27 |
| AHP (mV) | 6.7 ±1.0 | 6.6 ±1.2 | 0.09 | 0.93 |

Results of unpaired t tests and the respective *P* value is included for every corresponding measure. Part of the data described here is plotted on Supplementary Fig. S4.
*AP* action potential, *V resting* resting potential, *Rm* Membrane resistance, *AHP* afterhyperpolarization.

shown in Fig. 3 alone, we cannot definitively conclude that the potentiation of BLA synapses is sufficient to form the context-fear association. To circumvent this limitation, we developed a strategy to biochemically induce LTP specifically in the US neurons of the BLA by expressing the triple mutant CamKIIα T286D-T305A-T306A (CK2-DAA), which has been shown to induce LTP in hippocampal organotypic slices[24]. Using a similar approach to that described in Fig. 2, we bilaterally injected an AAV containing a CRE-dependent construct expressing CK2-DAA and a nuclear GFP marker into the BLA of FDC mice (Fig. 4A). We first tested whether CK2-DAA could induce LTP in BLA neurons in the absence of shock. We tagged neurons during a novel box exposure, and 7 days later, prepared slices for whole-cell recordings (Fig. 4B). Indeed, the CK2-DAA⁺ (GFP⁺) neurons exhibited an increase in the AMPAR/NMDAR ratio compared to the neighboring GFP⁻ neurons (Fig. 4C), without affecting their intrinsic excitability (Table 2). Of note, this increase in the AMPAR/NMDAR ratio was not seen with the overexpression of WT-CaMKIIα (Fig. 4C). Next, we injected FDC mice with the CK2-DAA virus in the BLA and used a similar unpaired training protocol to the one described in Fig. 3, in which CS and US were tagged independently on different days (Fig. 4D, E). CK2-DAA expression was minimal one day after TMP induction, but became substantial at 7 days (Supplementary Fig. 3F–H). This time, we found that the potentiation of BLA neurons tagged during US exposure alone was sufficient to drive both tone and context-mediated aversive associations, and this response generalized to multiple contexts (Fig. 4F–H). Notably, despite enhanced cue-induced freezing levels, no significant changes in locomotion or common anxiety behaviors were observed during elevated plus-maze, open field or marble burying tests (Supplementary Fig. 12A–C).

To rule out the possibility that any new fear memories were due to the formation of hybrid associations with the natural memory to box A[36], we included a control group in which the animals were habituated overnight to box A (ON Hab) before the exposure to the shock to prevent any conditioning[21] (Fig. 1C–E). Under these conditions, we still observed a generalized context fear response (Fig. 4F–H, purple line), even though the freezing to box A was reduced to a level similar to untrained boxes B and C (Supplementary Fig. 12D). Staining for cfos induction during exposure to box C showed that CK2-DAA expression promotes the reactivation of neurons that were tagged during US exposure (GFP⁺) relative to neurons tagged with a WT-CK2 control (Fig. 4I, J).

Together the results presented here suggest that induction of synaptic potentiation in CS- and US-responsive neurons is necessary to create a de novo, and specific, context fear association. However, when a biochemical method was used, the potentiation of US neurons alone was sufficient to create a non-specific fear response that generalized across multiple contexts.

This might be the result of a prolonged, non-specific, potentiation of presynaptic inputs to US-tagged neurons in the BLA that can drive generalized fear responses[28].

## Discussion

The results presented here, demonstrate that the essential underlying cellular plasticity for forming specific context fear associations is the potentiation of synaptic inputs onto US responsive neurons in the BLA, which in turn project selectively to the CeA and the fear output circuit[37]. This suggests that even for hippocampal-dependent tasks such as CFC, which involve the integration of multimodal sensory cues, the fear association takes place in the amygdala and the specificity of the response is likely determined by the nature of the inputs activated during learning and retrieval. In fear conditioning, it is the aversive shock that is actually "recalled", as the animals are present in the identical context box during both learning and retrieval. According to this view, the role of hippocampal and cortical structures is to respond to the context cues by recruiting the same neuronal ensembles from the original context exposure during learning. Fear is then expressed by the increased synaptic plasticity of BLA inputs and the subsequent activation of the US-linked fear circuit.

Using a cellular tagging approach to study learning-induced plasticity, we were able to demonstrate that context extinction reverses the synaptic potentiation of BLA neurons established during the original learning (Supplementary Fig. 2). Interestingly, a recent study demonstrated that tone-fear conditioning and extinction, bi-directionally modulate spine morphology in activated ensemble neurons within the auditory cortex-lateral amygdala circuit[13], which may be directly related to changes in synaptic potentiation observed here. Nevertheless, our findings do not rule out the possibility of extinction-specific synaptic plasticity occurring elsewhere in the circuit, as demonstrated by other studies[38,39]. Instead, our work suggests that extinction might involve not only new learning but also unlearning of the original memory. Understanding how these mechanisms cooperate during extinction, and identifying parameters that promote unlearning, such as the optimal time window for this process[40], could provide valuable insights for the reduction of relapse after exposure therapy in a clinical setting[41].

Previous lesion and pharmacological manipulations of various hippocampal and cortical structures have been shown to reduce context fear conditioning, likely reflecting altered processing of contextual information preventing reactivation of the appropriate BLA inputs. Since our approach involved delivery of both the CS and US in an unpaired manner to tag appropriate ensembles, it is possible that these stimuli individually produce plasticity at other sites within the circuit that contribute to the learning. Nevertheless, the plasticity critical for forming the Pavlovian association seems to be localized in the BLA. Further studies could take advantage of the tagging strategy described in this work to dissect the precise synaptic mechanisms behind the CS-US association. This could include investigating the roles of homosynaptic and heterosynaptic plasticity[42–44], identifying the subtypes of AMPARs involved[40], among others.

Synaptic plasticity observed in the hippocampus and other cortical regions during fear conditioning might be more involved in context encoding, such as creating more reliable ensemble reactivation upon context re-exposure. This hippocampal/cortical plasticity is likely to be particularly important in tasks that involve subtle context discrimination or complex spatial navigation, rather than simple cue recognition necessary for fear conditioning.

Our findings suggest that learning-activated BLA ensemble neurons form part of a brain-wide circuit that undergoes structural changes to store memories, often referred to as an engram. Engram neurons are active during learning, reactivated during retrieval, and essential for memory recall[8]. Our study further demonstrates that synaptic plasticity underlies these properties, as BLA neurons activated during learning and expressing immediate early genes: (1) undergo specific synaptic changes at learning, (2) are reactivated during retrieval, and (3) require these synaptic modifications for memory recall.

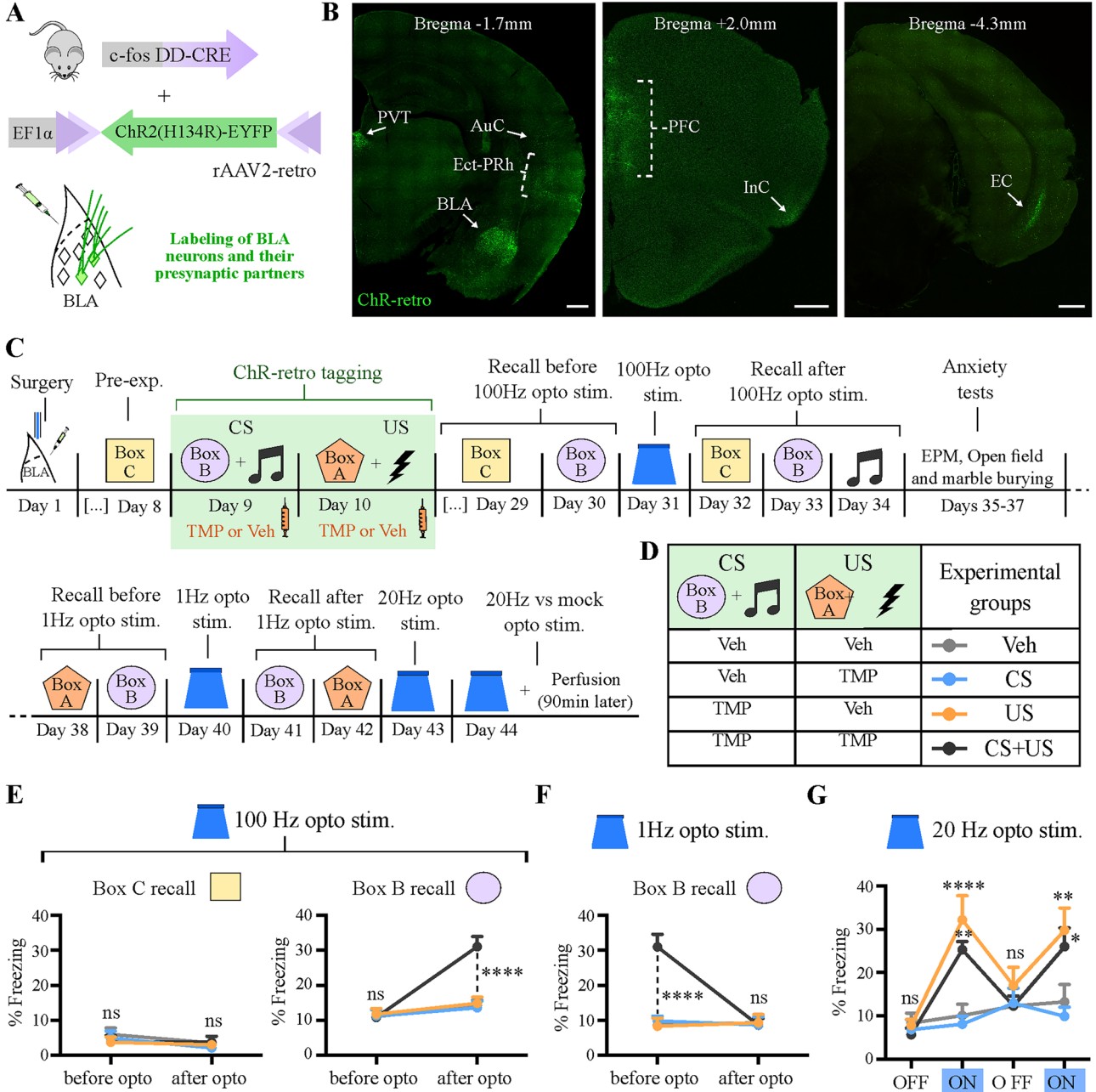

**Fig. 3 | 100 Hz optogenetic stimulation of both CS- and US- tagged neurons in BLA and their inputs is sufficient to create a de novo contextual aversive memory.** **A** AAV2-retro carrying a DIO-ChR2(H134R)-eYFP (ChR-retro) construct was bilaterally injected in the BLA of FDC mice, leading to expression of ChR-retro in tagged BLA local and input neurons. **B** Coronal sections showing the expression pattern of ChR-retro after injection in BLA and tagging during exposure to both CS and US (see **C, D**). The brain regions with the most significant expression are labeled. AuC Auditory Cortex (low expression), BLA Basolateral amygdala, EC Entorhinal cortex, including lateral and medial regions, Ect-PRh Ectorhinal and Perirhinal cortices, InC Insular cortex, PFC Prefrontal cortex, including prelimbic, infralimbic and mediorbital regions, PVT Paraventricular nucleus of thalamus. Scale bar: 500 mm. **C** Experimental design for optogenetic-mediated de novo memory generation and reversal. **D** Experimental groups and their respective tagging regimes, with either TMP or Veh IP injected 20 min after each behavior exposure. $N = 8$ mice per group. **E** 100 Hz optogenetic stimulation of BLA local and input neurons generates an artificial memory that is specific to the tagged context (Box B) only when both CS and US are tagged. **F** 1 Hz optogenetic stimulation reverses the artificially created memory to box B. **G** Freezing behavior during 20 Hz light stimulation occurs only when the US is tagged. Opto stim.: Optogenetic stimulation. ****$P < 0.0001$; **$P < 0.01$; *$P < 0.05$, ns not significant. Two-way RM ANOVA with Tukey test. Graph bars show mean $+/-$SEM.

## Methods

### Animals

All animal procedures were conducted in accordance with guidelines and protocols approved by the Institutional Animal Care and Use Committee at The Scripps Research Institute and the University of California San Diego. All mice were bred in house on a C57BL/6NTac background (purchased from Taconic Biosciences), genotyped, housed 2–5 per cage, and maintained on a 12 h-light/ 12 h-dark cycle with water and food ad libitum. Mice were 12-20 weeks old, in balanced male and female groups, as no significant difference in behavior were found among sexes.

The cfos-shEGFP transgenic mouse line (Fig. 1A), which expresses a short half-life EGFP (2 hours) under the control of the cfos promoter, was obtained as described previously[20], and it is available from The Jackson Laboratory, stock # 018306.

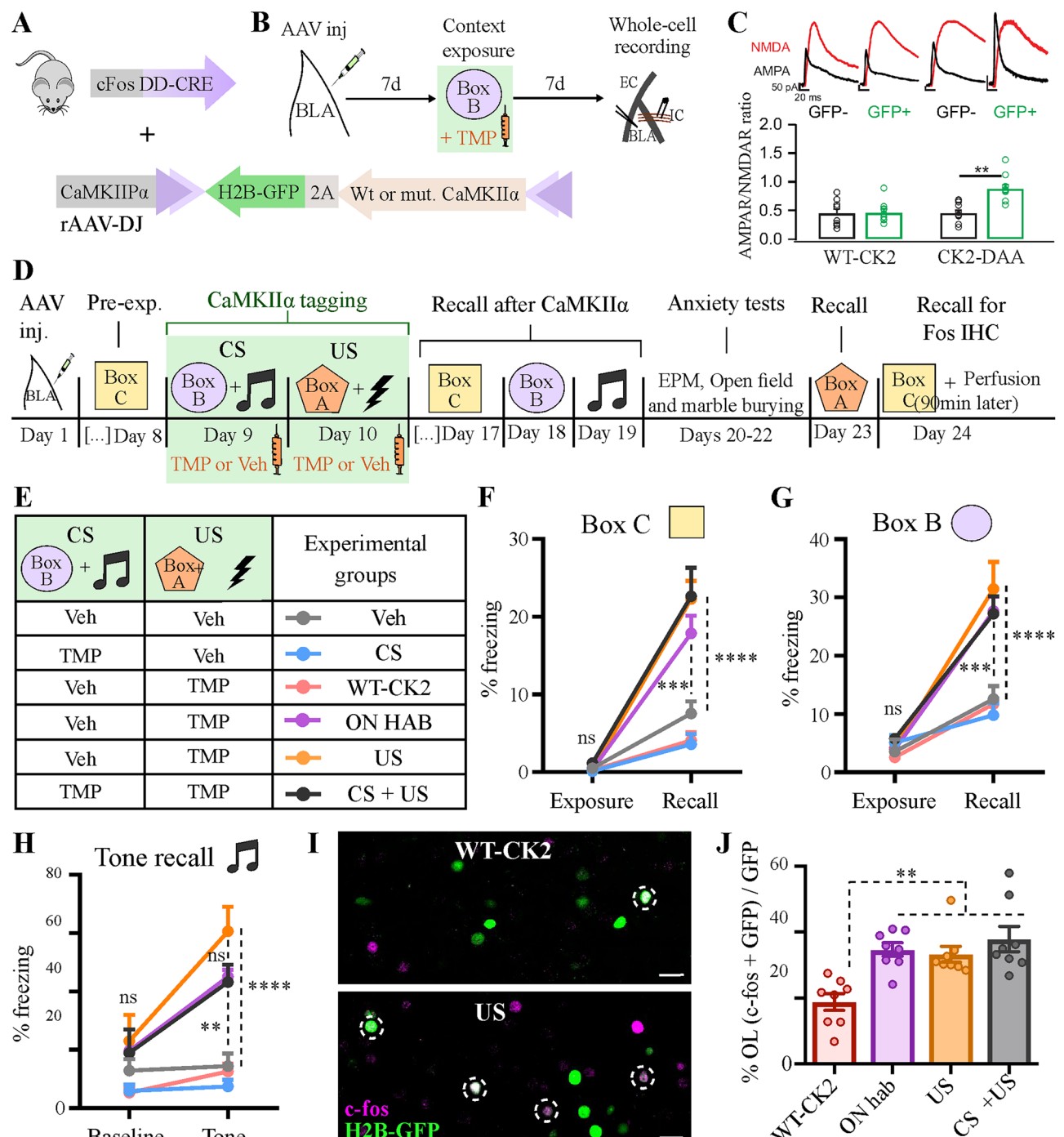

**Fig. 4 | Biochemical induction of synaptic potentiation in US + BLA neurons is sufficient to generate long-term aversive memories. A** rAAV-DJ carrying a DIO-CK2-2A-H2BGFP construct was injected bilaterally in the BLA of FDC mice. **B** Experimental design to test whether CK2-DAA induces synaptic potentiation in the BLA in the absence of US exposure. **C** CK2-DAA expression induces increase in AMPAR/NMDAR ratio, whereas WT-CK2 overexpression does not affect it. **D** Experimental design to test whether CK2-DAA mediated-potentiation creates a de novo memory association. **E** Experimental groups and their respective tagging regimes, with either TMP or Veh IP injected 20 min after each behavior exposure.

Mice in the ON hab group were kept overnight in box A before shock exposure to prevent any conditioning (Fig. 1C). $N = 8$ mice per group, except in the Veh group, where $N = 7$. **F–H** CK2-DAA potentiation of US neurons led to indiscriminate fear associations, including the untagged box C (**F**). **I, J** Representative coronal BLA sections showing overlap between activity markers during exposure (H2B-GFP) and recall to the untagged box C [(cfos IHC), (**I**)], quantified in (**J**). OL overlap. Scale bar: 20 mm. \*\*$P < 0.01$, \*\*\*$P < 0.001$, \*\*\*\*$P < 0.0001$, ns not significant; unpaired t-test (**C**), two-way RM ANOVA with Tukey test [(**F**)–(**H**)] or one-way ANOVA with Tukey test (**J**). Graphs show mean +/− SEM.

The double transgenic Arc-tTA/TetO-H2BGFP mouse line (Fig. 1F) was obtained by crossing the Arc-tTA line, in which the tetracycline transactivator (tTA) was knocked-in to the 5' exon of the Arc gene, with the TetO-H2BGFP line, which expresses a GFP fused to the human histone 1, H2bj, protein under the control of the tetO promoter (The Jackson Laboratory, stock # 005104). The histone-bound GFP is very stable and lasts for weeks[26]. This mouse line was based on the TetTag system[45], in which the activity-dependent gene (in this case, Arc) controls the expression of tTA, which binds to the tetO promoter, driving the expression of H2B-GFP. Another mouse line that was based on the TetTag system was the double

**Table 2 | CaMKIIα-T286D/T305A/T306A (CK2-DAA) expression in cfos⁺ BLA neurons tagged during novel box exposure does not affect their intrinsic excitability**

| | CK2-DAA⁻<br>n = 15 | CK2-DAA⁺<br>n = 9 | t | p |
|---|---|---|---|---|
| AP threshold (mV) | −47.1 ±1.3 | −47.0 ±2.1 | 0.06 | 0.95 |
| AP amplitude (mV) | 73.9 ±2.9 | 79.2 ±1.4 | 1.28 | 0.21 |
| AP half width (ms) | 0.82 ±0.04 | 0.87 ±0.06 | 0.77 | 0.45 |
| V resting (mV) | −78.9 ±1.8 | −74.5 ±2.0 | 1.55 | 0.14 |
| Rm (Mohm) | 167.9 ±12.9 | 185.9 ±36.8 | 0.55 | 0.59 |
| AP numbers at 300pA | 3.2 ±0.3 | 3.7 ±0.5 | 0.84 | 0.41 |
| Latency to spike (ms) | 6.5 ±0.8 | 7.7 ±1.6 | 0.75 | 0.46 |
| | n = 9 | n = 7 | | |
| AHP (mV) | 7.1 ±0.5 | 5.8 ±0.8 | 1.33 | 0.21 |

Results of unpaired t tests and the respective *P* value is included for every corresponding measure. *AP* action potential, *V resting* resting potential, *Rm* Membrane resistance, *AHP* afterhyperpolarization.

transgenic cfos-tTA/tetO-ChEF mice (Supplementary Fig. 9) generated by crossing single transgenic cfos-tTA[20] and tetO-ChEF-tdTomato mice[9]. For behavior experiments, single transgenic littermates were used for control. The Channelrhodopsin variant ChEF, which is more reliable at higher frequencies than standard variants 44, is expressed in cfos⁺ neurons during a time window determined by the absence of the drug doxycycline (Dox), to drive ChEF-TdTomato expression. In order to restrict activity-dependent labeling to targeted behavioral episodes, mice were raised from weaning on food containing Dox (On Dox, at 40 mg/kg). Four days prior to experimental manipulation (Three days for Arc-tTA/tetO-H2BGFP mice), mice were switched to food without Dox (Off-Dox) to allow the clearance of the drug before tagging. Mice were put back On Dox food 6 h after behavioral training, first at a concentration of 200 mg/kg to quickly stop H2B-GFP expression, and after 12 h, back to 40 mg/kg for maintenance for the remainder of the experiment.

The cfos-DD-Cre mouse (FDC, Figs. 2–4), was generated by a knock-in of the DD-Cre fusion protein in the cfos gene[23]. This mouse line expresses a Cre-recombinase fused to a destabilizing domain (DD), which leads to protein degradation[25]. The drug trimethoprim lactate (TMP) stabilizes the complex, providing a window of functional Cre expression. TMP peaks in the brain in 10 min and it is rapidly cleared, limiting the window of Cre expression to a few hours[23,25].

### Trimethoprim preparation and injection
TMP was purchased from Chem-Impex Int´l (# 03552), Inc in a lyophilized form and dissolved in H2O to a final concentration of 20 mg/ml. Immediately before injections, this solution was mixed with 2X phosphate-buffered saline (PBS) to a final concentration of 10 mg/ml. Each animal was injected with 150 µg/g for TMP to reach peak concentration in the brain[23,25]. Intraperitoneal injections (IP) were given 20-30 min after behavioral training, to minimize interference in conditioning and maximize labeling at a time of peak cfos expression[23].

### Viral constructs
All rAAVs were made in house with the DJ serotype[46], except for ChR-retro (pAAV-EF1α-DIO-hChR2(H134R)-EYFP), which was acquired from Addgene (plasmid # 20298) with the rAAV2-retro serotype[27] to target BLA and its projection neurons (Fig. 3).

The rAAV-DJ production was carried out following protocol adapted from previous work[47]. With the exception of Syn-GFP (see below), all rAAVs made in house contain the CaMKIIα minimal promoter (CK2min), favoring viral expression in excitatory neurons.

The recombinant AAV vectors used for viral production were:
- DIO-WT-CK2: pAAV-CK2min-DIO-CaMKIIα-2A-H2BGFP
- DIO-CK2-D: pAAV-CK2min-DIO-CaMKIIα-T286D-2A-H2BGFP

- DIO-CK2-DAA: pAAV-CK2min-DIO-CaMKIIα-T286D/T305A/T306A-2A-H2BGFP
- Timer: pAAV-CK2min-H2B-Timer
- DIO-Timer: pAAV-CK2min-DIO-H2B-Timer
- DIO-Syn-Venus: pAAV-Ef1α-DIO-Synaptophysin-Venus (provided by Anton Maximov's lab)

AAVs made in house were tittered through qPCR. The AAV vectors were injected in the BLA with the following titers*#:
- DIO-ChR-retro: $1.0 \times 10^{13}$ units/ml genome copies (GC)/ml
- DIO-WT-CK2, DIO-CK2-D and DIO-CK2-DAA: $4 \times 10^{12}$ GC/ml
- Timer: $8 \times 10^{12}$ GC/ml
- DIO-Timer: $4.5 \times 10^{13}$ GC/ml
- DIO-Syn-Venus: $1.1 \times 10^{13}$ GC/ml

*: A minute volume (~1:100) of the dye Fast Green FCF (Tokyo Chemical Industry, # F0009) diluted at 1 mg/ml in H2O was mixed to the virus aliquot before injection to aid in visualization during surgery.

#: Whenever a combination of viruses was used (e.g. Supplementary Fig. 3), their concentrations were adjusted to match (1:1), and further dilutions were avoided.

### Surgical procedures
For all surgeries, mice were first anesthetized with 5% isoflurane/oxygen mix in an induction chamber ($20 \times 10 \times 8$ cm$^3$) until proper sedation was achieved. Animals were transferred to a stereotaxic apparatus (Kopf Instruments, Model # 1900), and anesthesia was maintained with 1.5–2% isoflurane/oxygen mix. Hair was removed from the head and the skin disinfected with 70% ethanol followed by the application of 9% Povidone-iodine solution (Betadine) and again 70% ethanol to remove excess of Betadine. A small incision was then made on the scalp and the skin was carefully retracted to allow direct access to the skull. Body temperature was maintained using a deltaphase isothermal pad (Brain Tree Scientific, Inc.). Veterinarian petrolatum ointment (Puralube) was used to prevent eyes from drying during surgery. After the surgery, a subcutaneous injection of 2.5 µg/g flunixin (Sigma-Aldrich, # 33586) was given to reduce post-surgical pain and inflammation. After surgery, mice were allowed to recover for 7 days before any further experimental procedure.

AAV injection: Small holes were drilled to the skull, and the appropriate AAV was delivered bilaterally (unless otherwise specified) to BLA using a thin glass needle (Sutter Instrument, #BF100-50-10) of 30–40 µm tip diameter at the following Bregma coordinates: −1.45 mm anteroposterior (AP), ±3.32 mm mediolateral (ML), −4.9 mm dorso-ventral (DV). The definition of Bregma was the (often imaginary) intersection between the sagittal suture and the curved formed by the coronal suture, similar to previously described in rats[48]. Injection volume was 300 nl per hemisphere, enough to achieve 70–100% coverage of BLA, except for ChR-retro and Syn-Venus, in which 200 nl was injected to avoid any off-target labeling, achieving 50–70% coverage of BLA. The injection speed was kept at 100 nl/min using an infusion pump (Pump 11 Elite Series, Harvard Apparatus) connected to the glass needle through a 2.0 µl syringe (Hamilton Company, #88500) and hard plastic tubings filled with Mineral Oil (Fisher Scientific, BP26291). The glass needle was left in place 3 min prior to injection and 5 min following infusion to minimize backflow. At the end of the surgery, if no optic fiber placement was required, the skin was sutured back in place with sterile absorbable surgical suture (Ethicon, # J844G).

Optic fiber placement: Prior to optic fiber placement, two screws (Plastics One, # 00-96×1-16) were added to the skull to provide extra support and stability at −3.5 mm AP, −1.5 mm ML and −0.85 mm AP, −1.3 mm ML. 200 µm-core optic fibers (NA 0.37) connected to ceramic ferrules (Newdoon, FOC-C-200-1.25-0.37-5.0), were placed at −1.45 mm AP, ±3.32 mm ML, −4.6 mm DV (300 µm above the center of AAV injection in the case of ChR-retro, Fig. 3). Dental cement (Stoelting, # 51458) was then added to secure the optic fibers in place, forming a head cap.

## Slice electrophysiology

Following training at specified time points, mice were anesthetized with isoflurane drops in a glass jar followed by decapitation. Brains were quickly removed and placed in ice-cold N-methyl-D-glucamine (NMDG) based recovery solution containing (in mM): NMDG 93, HCl 93, KCl 2.5, NaH2PO4 1.2, NaHCO3 30, HEPES 20, Glucose 25, sodium ascorbate 5, thiourea 2, sodium pyruvate 3, MgSO4 10, CaCl2 0.5, bubbled with carbogen gas (95% $O_2$/5% $CO_2$)[49]. Coronal slices (300 μm) were cut in a vibratome (Compresstome VF300) and incubated in the same NMDG solution for 10–15 min at 32 °C before being transferred to a HEPES aCSF solution containing (in mM): NaCl 92, KCl 2.5, NaH2PO4 1.2, NaHCO3 30, HEPES 20, Glucose 25, sodium ascorbate 5, thiourea 2, sodium pyruvate 3, MgSO4 2, CaCl2 2 at room temperature[49], bubbled with carbogen gas, for an additional hour before recording. Recording was conducted in an open bath chamber continuously perfused with warm (30–32 °C) aCSF containing (in mM): NaCl 124, KCl 2.5, NaH2PO4 1.2, NaHCO3 24, HEPES 5, Glucose 12.5, MgSO4 2, CaCl2 2, at the rate of 2–3 mL/min, bubbled with carbogen gas. Olympus BX51WI upright microscope (Scientifica) was used for viewing the slices in differential interference contrast with 60× magnification. Identification of GFP$^+$ cells was conducted with a 470 nm LED (CooLED) and a CCD camera (ORCA-Flash4.0LT, Hamamatsu).

Recordings of BLA pyramidal cells were performed with glass micropipettes (resistance 3.5–5 MΩ) pulled by a PC-10 puller (Narishige) and filled with intrapipette solution containing (in mM): 120 Caesium methansulphonate, 20 HEPEs, 0.4 EGTA, 2.8 NaCl, 5 tetraethylammonium chloride, 2.5 MgATP, 0.25 NaGTP (pH 7.3, 285 mOsm)[17]. 2 mM Alexa Fluor 594 was added to the intrapipette solution for a subset of cells. For measuring AMPAR/NMDAR ratio, brain slices were perfused with a γ-aminobutyric receptor A (GABAA) antagonist picrotoxin (100 μM, Tocris). A concentric bipolar stimulation electrode (FHC) was placed in the internal capsule fibers projecting to the basolateral amygdala. Due to the lack of information in how contextual information reaches the amygdala, we decided to stimulate the internal capsule, as it has been extensively used in tone conditioning, and therefore could serve as a positive control in such behavioral condition. Stimulation intensity ranged from 50 to 200 μA. Cells were voltage clamped at +40 mV and evoked EPSCs were recorded every 10 s. An 80% maximum response was used for baseline recording over 5 minutes to ensure stability. Once a stable baseline of EPSCs (compound AMPAR + NMDAR current) was established, an NMDAR antagonist AP5 (50 μM, Tocris) was applied for 5 min and AMPAR EPSCs were recorded in the presence of AP5. NMDAR current was obtained by subtracting the AMPAR EPSC from the compound EPSC. For cells patched from the same slice, the washout time exceeded 1 h to ensure complete removal. Average of 10–15 EPSC traces were used. Paired pulsed ratio (PPR) of evoked EPSC amplitudes was measured following two stimulations separated by 50 msec.

For measuring intrinsic properties and excitability, intrapipette solution containing (in mM): 123 K-gluconate; 2 MgCl2; 8 KCl; 0.2 EGTA; 10 HEPES; 4 Na2-ATP; 0.3 Na-GTP was used. Resting membrane potential was measured at current clamp when whole-cell patch was established. Depolarization currents of increasing amplitude (10 pA steps, 250 msec) were injected into the cell through the patch-pipette in current clamp mode. The action potential (AP) evoked by the smallest current injection was used for measuring threshold, amplitude, and half width. Input resistance was estimated in the linear portion (−50 pA to 50 pA) of the current-voltage plots. Repetitive spiking was measured using a series of 250 msec current steps (−50 pA to 250 pA, 50 pA/step). The input-output relationship between the current size and action potential numbers was constructed and the latency to the 1st spike at 250 pA current injection was measured. Post-burst afterhyperpolarization (AHP) was measured at −5 mV below the AP threshold and AHP was evoked using a 100 msec current step that elicits 4 APs[50]. For mEPSCs, cells were held at –70 mV in voltage-clamp mode and recorded in the presence of tetrodotoxin (0.5 μM) and picrotoxin (100 μM; Tocris) for 5 min. Cells which access resistance were >20 MΩ or that varied more than 30% during the recording were excluded from analysis. Each group contained 5–9 cells from 4 to 6 mice.

In order to minimize experimental variation in the recordings, we included only neighboring tagged and untagged neurons from the same slice preparation in the analysis, although no substantial difference was found in the excluded data pool. In addition, the recordings were conducted by an experimenter blind to the behavioral group. Multiclamp 700B amplifier and pClamp10 software were used for data acquisition (filtered with 2 kHz low pass filter) and digitization (10 kHz sampling frequency). Electrophysiological data were analyzed by Clampfit, Mini Analysis and Igor.

## Handling

All behavioral experiments were conducted during the facility light cycle (6:00 am to 6:00 pm). All mice were individually habituated to the investigator by handling for one minute on each of five consecutive days. Handling took place in the mouse colony. Immediately prior to each handling session, mice were transported by wheeled cart to and from the vicinity of the experimental context rooms to habituate them to the journey.

## Fear conditioning

Tone and Context conditioning were carried out in MedAssociates boxes (box A, see below). Unless otherwise specified, each session lasted for 6 min, starting with 3 min acclimation to the context, followed by three 30-s-long tones (5 kHz, 70 dB) starting at 180 s, 240 s and 310 s, that were co-terminated with 1 s 0.6 mA foot shocks.

For context conditioning only (e.g. Fig. 2J, and during US tagging in Figs. 3, 4), the same 6-min protocol was used, but no auditory tones were played. 1 s 0.6 mA foot shocks were delivered at 209 s, 269 s and 339 s. For CK-DAA US tagging (Fig. 4), 1.5 mA foot shocks were used to promote labeling of US-tagged neurons.

For context conditioning with unpaired tones (e.g. Fig. 1C), the same auditory tones (30 s, 5 kHz, 70 dB) and foot shocks (1 s, 0.6 mA) were given, but following a different schedule: tones started at 100 s, 150 s, and 210 s and foot shocks were given at 270 s, 300 s and 340 s.

For overnight habituation (ON), animals were introduced to the box 16 h before shock exposure, 2 h before the end of the light cycle. Food and water (hydrogel) were provided, and the 12 h-light/ 12 h-dark cycle was followed. On the next day, 2 h after the beginning of the light cycle, shocks were delivered as for context contitioning. The long habituation to the box inhibits conditioning with foot shock exposure[21].

Box exposure lasted for 6 min, with no shock or tone, unless otherwise specified. Tone exposure in Box B was played through an iPad calibrated with a sound level meter (MedAssociates, # ANL-929A-PC) using the Tone Generator Pro app. Auditory tones were delivered at the same time as in tone conditioning: three 30-s-long tones (5 kHz, 70 dB) starting at 180 s, 240 s and 310 s.

Context recall trials were 180 s long, while Tone recall trials were 220 s long, with a single tone (30 s, 5 kHz, 70 dB) starting at 180 s. Tone recall was conducted in a different context (box "D") to avoid context extinction.

Freezing behavior, defined as the absence of all movement except that required for respiration and heartbeat, was first manually counted for ~20% of animals in every experiment in order to adjust the threshold used in the automated system to give the final counts for all animals. The MedAssociates system was used for box A and tone recall, and boxes B and C were recorded with a camera (Logitech C270 HD Webcam) mounted above them and subsequently analyzed using automated video tracking software (ANY-maze, Stoelting Co.).

Context extinction: For context extinction (Supplementary Fig. 2), animals were exposed to the conditioning box (box A) for 30 min/day during five consecutive days. The reported freezing during extinction trials consists of the first 3 min of each session. Retraining was carried out using the same protocol as for novel context conditioning.

Contexts environments: Box A, B and C were located in different rooms, with different illumination patterns to help with discrimination among contexts.

Box A consisted of a rectangular box (25.5 × 29 × 25.5 cm) with metal rod grid floor, two metal sidewalls, one black and white checkerboard back

wall, and transparent Plexiglas front wall and ceiling. A piece of paper with peppermint odor was kept at the vicinity of the box to provide a distinct odor cue. A direct fluorescent-white light illuminated the box and the room lights were on. Box A exposures were carried out inside the MedAssociates chamber. The box was cleaned with 70% ethanol between sessions. All conditioning and foot shock exposures were carried out in Box A.

Box B consisted of a white acrylic square box ($25.5 \times 25.5 \times 25.5$ cm) open on top. A piece of paper with lavender odor was kept at the vicinity of the box to provide a distinct odor cue. The box was cleaned with 70% ethanol between sessions. An indirect fluorescent-white light illuminated the box and the room lights were on.

Box C consisted of a gray opaque plastic box ($40 \times 20 \times 25$ cm) open on top. A piece of paper with orange odor was kept at the vicinity of the box to provide a distinct odor cue. The box was cleaned with 70% ethanol between sessions. An indirect fluorescent-white light illuminated the box and the room was dimly illuminated.

Tone recall was carried out in the Med Associates chamber. To make it distinct from box A, the whole chamber ($25.5 \times 29 \times 25.5$ cm) was covered in white plastic walls and floor, except for the ceiling and the transparent Plexigas front wall. Mice were put inside a smaller transparent plastic box ($19 \times 17.8 \times 17.8$ cm) with fresh bedding on the floor, and this box was inserted into the bigger chamber. A piece of paper with cinnamon odor was kept at the vicinity of the box to provide a distinct odor cue. The box was cleaned with 70% isopropanol and the bedding was replaced between sessions. A direct fluorescent-white light illuminated the box. The room was dimly illuminated, and a black curtain was used to make it seem smaller.

### Anxiety tests

The anxiety tests were carried out on consecutive days, starting with the elevated plus-maze, the open field test, and finishing with the marble burying test.

Elevated plus-maze (EPM): Mice were tested in an EPM that was elevated 75 cm from the floor by 6 legs, following a standard protocol[51]. Each arm measured $30 \times 5$ cm and closed arms had walls of 30 cm. Animals were always placed at the center of the maze, facing the open arm opposite to where the experimenter was, and allowed to explore for 5 min. The maze was cleaned with 70% ethanol before each session. Behavior was recorded using a camera mounted above the arena (Logitech C270 HD Webcam). Animal movements and position were subsequently analyzed using automated video tracking software (ANY-maze, Stoelting Co.) to determine freezing (see above), total distance traveled, and time in open areas (center + open arms).

Open field test: Mice were placed in the center of an open arena (50 cm $\times$ 50 cm) and allowed to explore for 5 min. A piece of paper with vanilla odor was kept in the vicinity of the box to provide a distinct odor cue. The arena was cleaned with 70% isopropanol before each session. Behavior was recorded using a camera mounted above the arena (Logitech C270 HD Webcam). Animal movements and position were subsequently analyzed using automated video tracking software (ANY-maze, Stoelting Co.) to determine freezing (see above), the distance traveled, and time spent at the center or edges of the open arena (center defined as square of $30 \times 30$ cm).

Marble burying test: The test was conducted as described elsewhere[52]. Briefly, mouse cages were filled approximately 5 cm deep with wood chip bedding, and 10 multi-colored opaque glass marbles were evenly arranged on top of the bedding. A single mouse was carefully put in each cage, not on top of any marble, and the cage was closed. Each test session lasted for 30 min and buried marbles (threshold was 2/3 buried) were manually counted. The duration of the test was determined by preliminary experiments with wild-type mice that had been fear conditioned; 30 min was the average time required to bury half of the marbles.

### Optogenetic stimulation

All optogenetic stimulation sessions were carried out using a 473 nm diode-pumped solid-state (DPSS) laser (Laser Century). Blue light was delivered bilaterally to the BLA through 200 µm-core optic fibers (NA 0.37) connected to ceramic ferrules (Newdoon, FOC-C-200-1.25-0.37-5.0). Laser intensity at the tip of the fiber was set to 10 mW.

100 Hz optogenetic stimulation: Our protocol was based on what has been previously established to induce LTP in auditory inputs to LA[15,30]. Mice were attached to the optic fiber patch cords and allowed to individually explore an open mouse cage with fresh bedding for 5 min. Optogenetic stimulation consisted of 5 trials of light with 3 min inter-trial interval. Each trial consisted of 2 ms, 100 Hz pulsed light with a total duration of 1 s (20% duty cycle). 3 min after the end of light stimulation, the patch cord was detached, and the mouse was returned to its home cage.

1 Hz optogenetic stimulation: Our protocol was based on what has been previously established to induce LTD in auditory inputs to LA[14,15,30]. Mice were attached to the optic fiber patch cords and allowed to individually explore an open mouse cage with fresh bedding for 5 min. Optogenetic stimulation consisted of 900 pulses of light, each 2 ms, at 1 Hz (0.2% duty cycle). 3 min after the end of light stimulation, the patch cord was detached, and the mouse was returned to its home cage.

20 Hz optogenetic stimulation: Our protocol was based on what has been previously reported for driving freezing behavior[12], using parameters tested in the BLA 9. Mice were attached to optic fiber patch cords and placed at the center of a novel circular white plastic box with 25 cm of diameter and 20 cm high walls. A piece of paper with ginger odor was kept at the vicinity of the box to provide a distinct odor cue. The protocol started with 3 min of baseline, followed by 3 min of Light ON, 3 min Light OFF and 3 min Light ON, for a total of 12 min. The Light ON periods consisted of Light pulses at 20 Hz, with 10 ms Pulse width (20% duty cycle). A mock 20 Hz optogenetic stimulation consisted of 12 min exploration, with no periods of light ON.

### cfos immnohistochemistry

90 min following the last behavioral session, mice were overdosed with isoflurane and transcardially perfused with 1x PBS followed by 4% paraformaldehyde (PFA) in 1x PBS. Brains were removed and kept in 4% PFA 1x PBS at 4 °C overnight. On the following day, brains were washed in 1x PBS and sliced at 75 µm using a Leica VT100S vibratome. Immunohistochemistry was performed in free floating slices, and all the incubations were performed in horizontal shaker at gentle speed. Brain slices were first incubated for 3 h in blocking solution (10% normal goat serum [Jackson Immuno Research, 005-000-121], 0.2% Triton X-100, 0.05% Sodium azide) at room temperature. Following this step, slices were incubated for 12–16 h in blocking solution containing 1:700 rabbit anti-cfos antibody (Cell Signaling Technology, #2250S) at 4 °C. After three 10 min washes in 0.2% Triton 1x PBS solution, slices were incubated for 3, 4 h in blocking solution containing 1:750 goat anti-rabbit antibody conjugated with Alexa dye 647 (Invitrogen, #A-21244) at room temperature. After one wash in 0.2% Triton 1x PBS solution, slices were incubated for 30 min with 200 µM DAPI (Invitrogen) diluted in 0.2% Triton 1x PBS solution for counterstaining. Slices were then washed three times in 1x PBS and mounted in microscope slides (Fisherbrand Superfrost Plus, Fisher Scientific) with antifade mounting medium (SlowFade Gold, Invitrogen). Imaging was carried out within a week.

### Confocal microscopy and cell counting

All brains were fixed in 4% paraformaldehyde overnight, washed in 1x PBS and cut using a Leica VT100S vibratome (50–75 µm slices). Slices were counterstained with DAPI (Invitrogen) and mounted on microscope slides (Fisherbrand Superfrost Plus, Fisher Scientific) with antifade mounting medium (SlowFade Gold, Invitrogen). Some slides were counterstained with Hoechst 33258 (Invitrogen), which facilitated BLA visualization from surrounding amygdalar nuclei but make it more difficult for automated cell counting due to staining of fibers.

Images were acquired using a Nikon A1 R+ confocal microscope with 4X (0.2 NA, overview images) and 20X (0.75 NA, detailed images for cell counting) plan apochromat objective lens and lasers at 405 nm, 488 nm, 561 nm and 640 nm. All acquisition parameters were kept constant within

each magnification for a given experiment. All cell counting experiments were conducted blind to experimental group, using 3-4 coronal slices per region of interest for each mouse (slices with the most viral labeling were chosen). Cell counts and overlap quantification were performed using an ImageJ macro which identifies only cells that overlap with DAPI (available upon request). The reactivation of neurons tagged with CaMKIIα AAVs (GFP$^+$), which has been shown to correlate with memory retrieval (Reijmers et al., 2007; Tayler et al., 2013), was calculated as the number of cells that are both cfos$^+$ (IHC) and GFP$^+$, divided by the total number of GFP$^+$ cells (Figs. 2I and 4J).

For synaptophysin-Venus (Syn-Venus) AAV experiments (Supplementary Fig. S10), regions of interest were drawn around the CeA, NAcc or PFC in ImageJ, and the total area was measured. Next, all images were thresholded to include only the brightest signal (by eye), and the thresholded signal area was measured. The signal area (Syn-Venus)/total area was averaged for each mouse before averaging each group. Such measurement was found to correlate well with the number of labeled axonal boutons counted in images with higher resolution[23].

### Statistics

Behavioral data analysis and statistics were conducted using Prism (Graphpad). Electrophysiological data analysis and statistics were conducted using Clampfit and Igor. Data from two groups were compared using two-tailed unpaired Student's $t$ tests. Multiple group comparisons were assessed using one-way or two-way repeated-measures (RM) ANOVA with post hoc tests as described in the appropriate figure legend. Data are presented as means ± the standard error of the mean (SEM). The number of mice is indicated by N, except when referring to the number of recorded cells. Statistical results are included in the figure legends.

### Reporting summary

Further information on research design is available in the Nature Portfolio Reporting Summary linked to this article.

### Data and materials availability

All data are available through the public repository Figshare and can be assessed via https://doi.org/10.6084/m9.figshare.28559645.v1.

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

## Acknowledgements
This work was supported by the National Institute of Health (grants R01MH057368 and R01DA035657, to M.M.) and by a Natural Sciences and Engineering Research Council of Canada discovery grant (RGPIN-2018-04401 to Q. Y.). L.M.C. was supported by the CAPES Foundation - Brazil (BEX 18818/12-9) and the Howard Hughes Medical Institute International Student Research Fellowship. A.F.S. was supported by the GABBA PhD program (FCT fellowship SFRH/BD/52037/2012). We thank Anton Maximov for providing the Syn-Venus AAV, Thomas Hnasko for providing optogenetics equipment and Kathryn Fife for technical assistance, Ian Winchester and Yena Lee for assistance with the mouse colony and genotyping, and Natasha Weaver for administrative support during the years at The Scripps Research Institute. We are also thankful to Roberto Malinow, Stefan Leutgeb, Nicholas Spitzer, Thomas Hnasko and Kiriana Cowansage for helpful discussion and suggestions.

## Author contributions
L.M.C. and M.M. conceived the idea behind this work. L.M.C., A.F.S. Q.Y. and M.M. designed the experiments. L.M.C. and Q.Y. performed electrophysiological experiments. Q.Y. analyzed electrophysiological data. L.M.C. performed surgeries, behavior experiments, imaging and analysis. B.C.D. performed imaging analysis. A.F.S. and S.R. performed behavior experiments. W.D. assisted with tracing experiment. N.J. provided technical support. N.J. and M.M. designed and produced AAVs. E.J.Y. assisted with histological procedures. L.M.C., A.F.S., Q.Y., and M.M. wrote the manuscript. B.C.D. and W.D. edited the manuscript and contributed critically to data interpretation and discussion.

## Competing interests
The authors declare no competing interests.
