## [Transparent Peer Review file · Communications Biology]

Synaptic Potentiation of Engram cells is Necessary and Sufficient for Context Fear Memory

Corresponding Author: Dr Qi Yuan

Version 0:

Reviewer comments:

Reviewer #1

(Remarks to the Author)

Yuan et al. demonstrated the causal relationship between contextual fear memory and synaptic changes in the engram cells of the basolateral amygdala (BLA). To this end, the authors combined optogenetic stimulation of the engram cells with the biochemical intervention of synaptic plasticity by expressing CaMKII mutants and examined their impacts on the behaviors and physiological properties of synaptic responses. Their conclusions are solid and well supported by their data and well-designed controls. I believe the manuscript is suitable for publication in its present form.

Reviewer #3

(Remarks to the Author)

Version 1:

Reviewer comments:

Reviewer #1

(Remarks to the Author)

In the revised manuscript by Cardozo et al the authors have sophisticated the manuscript by addressing the points, particularly the usage of key terminology, raised by Reviewer #2 in the first round of the review. My conclusion remains the same – the manuscript is suitable for publication in its present form.

I believe the authors' comment well addressed the major point raised by Reviewer #2, and the additional discussion paragraph strengthened their claim. I also conclude that the other minor points are also addressed, further improving the quality of the manuscript.

Reviewer #3

(Remarks to the Author)

I've reviewed the authors' responses, and they answered all my comments and modified the text accordingly. Congratulations to all authors for this beautiful work.

Reviewers' comments:

Reviewer #1 (Remarks to the Author):

Yuan et al. demonstrated the causal relationship between contextual fear memory and synaptic changes in the engram cells of the basolateral amygdala (BLA). To this end, the authors combined optogenetic stimulation of the engram cells with the biochemical intervention of synaptic plasticity by expressing CaMKII mutants and examined their impacts on the behaviors and physiological properties of synaptic responses. Their conclusions are solid and well supported by their data and well-designed controls. I believe the manuscript is suitable for publication in its present form.

We thank the reviewer for their detailed reading of the manuscript and appreciate their comments.

Reviewer #2 (Remarks to the Author):

This study by Cardozo et al. presents valuable insights into the nature and distribution of the synaptic changes of basolateral amygdala (BLA) neurons underlying fear learning and memory. The authors combined behavioral experiments with ex vivo physiology, biochemical methods and optogenetic techniques in transgenic mice to study the cellular and molecular mechanisms in BLA that underly the formation of contextual fear memories.

The manuscript is well-written, and the experiments are original for two main reasons: 1) most studies examine the nature of amygdala-based plasticity underlying tone-, but not context-fear associations, and 2) these studies do not disentangle the nature of the modifications between engram and non-engram cells. The authors found that learning-activated neurons (i.e., cFos-tagged neurons) in the BLA undergo long-lasting (up to 7 days) synaptic potentiation at the time of fear learning, which can be reversed by fear memory extinction. They further demonstrated that this synaptic potentiation is both necessary and sufficient to maintain a long-term association between a context and a foot shock. I find the presented work highly valuable for understanding the fundamental cellular mechanisms of learning and memory.

My main concern is the incorrect use of the term engram throughout the manuscript and the overstatement of tagging engram cells using the cFos-eGFP transgenic mouse tool. The authors imply that BLA GFP+ neurons (i.e., neurons activated by fear learning) are engram cells. This is inaccurate. Engram neurons are not merely those activated during a learning experience but are specifically those reactivated during memory retrieval (Reijmers et al., 2007; Tayler et al., 2013; but also Trouche et al., 2013). Examples of incorrect usage include statements such as “neurons active during learning and expressing the immediate early gene cFos, known as engram cells” or “we examined synaptic responses of engram neurons (GFP+/cFos+) and GFP- neurons”. However, in some instances, the authors appropriately refer to these cells as 'learning-activated neurons,' which is correct. This issue needs to be addressed comprehensively, with the terminology revised throughout the manuscript for accuracy.

We appreciate the reviewer's insightful comment and have updated the terms throughout the text accordingly. However, we would like to point out that to be classified as engram neurons, stringent criteria have been proposed (Tonegawa *et al.* 2015, *Neuron*; Mayford 2014, *Philosophical Transactions Royal Soc B Biological Sci*), including activation during learning, enduring cellular/biochemical changes (plasticity), reactivation during retrieval, and necessity and sufficiency for memory recall. Our findings demonstrate that learning-activated BLA ensemble neurons meet these criteria. Figure 1 shows their activation during learning (c-fos or arc expression), while Figures 2H-I illustrate their reactivation during retrieval, correlating with memory recall. Additionally, Figures 2 and 4 demonstrate that these neurons undergo synaptic plasticity that is necessary (Figure 2) and sufficient (Figure 4) for memory retrieval.

A final discussion further examines engram criteria and the specific contributions of our work (Page 7, line 312-318): **“Our findings suggest that learning-activated BLA ensemble neurons form part of a brain-wide circuit that undergoes structural changes to store memories, often referred to as an engram. Engram neurons are active during learning, reactivated during retrieval, and essential for memory recall (Tonegawa et al., 2015). Our study further demonstrates that synaptic plasticity underlies these properties, as BLA neurons activated during learning and expressing immediate early genes: (1) undergo specific synaptic changes at learning, (2) are reactivated during retrieval, and (3) require these synaptic modifications for memory recall.”**

Minor comments:

1) Methods: Some important information is missing regarding the ex vivo experiments. For instance, were the acute slices oxygenated?

Yes, all aCSF solutions were bubbled with carbogen gas (95% O₂/5% CO₂). This detail has now been included in the methods section (see Page 10, lines 445, 450, 453). We apologize for the oversight.

Why was a HEPES-ACSF solution used for recovery, given that extracellular HEPES inhibits GABA receptors?

We have added a reference (Ting *et al.*, 2014) for the use of HEPES-aCSF holding solution. The inclusion of 20 mM HEPES was intended to mitigate edema during incubation. The recording was carried out in aCSF without HEPES.

Additionally, information about the electrical stimulation protocol is lacking. How was the stimulation intensity determined? For example, was it adjusted to elicit a response with each stimulation, or was it set to 40% of the maximal response? This requires clarification.

We have added **“An 80% maximum response was used for baseline recording over 5 minutes to ensure stability”** (Page 11, lines 467-468).

Lastly, if multiple cells were recorded from the same slice (as suggested in Figure 1D), why was the more commonly employed protocol (Kim WB, Cho J-H, 2020) not used? This protocol measures AMPA responses at -70 mV and NMDA responses at +40 mV without the application

of AP5, thereby avoiding potential issues with drug washout. If AP5 was used, how was it ensured that the drug was completely washed out?

We followed the protocol of Namburi et al. (2015) but acknowledge that the Kim (2020) protocol without AP5 may have been more advantageous. We have added “**For cells patched from the same slice, the washout time exceeded 1 hour to ensure complete removal**” (Page 11, lines 471-472).

2) Figure 1 legend: The sentence “These images were taken from a slice of an Arc-tTA/TetO-H2B-GFP mouse” should be clarified. Could the authors provide images from the cFos-shEGFP mice instead, as their patch-clamp experiments shown in Figure 1E were conducted using this mouse line?

We aimed to demonstrate the procedure for confirming GFP⁺ versus GFP⁻ during whole-cell patching. Since the method is consistent across all transgenic mice, we selected the best example for imaging.

3) Figure 1—figure supplement 2E: The x-axis labeling (e.g., “1-2”) is unclear. Please provide a clearer representation.

We have now clarified this in the figure legend “Freezing levels across the extinction trials (**trial 1-5; x-axis**) and after retraining”

4) Figure 1F: Could the authors justify their decision to use Arc-tTA/TetO-H2BGFP mice instead of cFos-tTA/TetO-H2BGFP mice? The use of this new mouse line limits the possibility to directly compare their findings with previous results.

The Arc-tTA/TeO-H2BGFP mouse line was used instead of the cfos line due to animal availability at the time of the study. The percentage of GFP⁺ cells was comparable to that in the cfos-shEGFP transgenic mouse used previously (Figure 1—figure supplement 2, A and B), and synaptic potentiation, measured by AMPA/NMDA ratios, was similar in cfos⁺ and Arc⁺ neurons. These convergent results demonstrate that the observed effect is not due to some peculiarity of cells that express cfos but is more likely to be an intrinsic property of neurons potentially involved in memory encoding.

5) Figure 2B: The authors show an image of the posterior BLA, whereas fear engram neurons are preferentially located in the anterior BLA (Kim et al., 2016; Zhang et al., 2020). Could the authors provide additional data or discussion regarding potential differences in synaptic responses between anterior and posterior tagged BLA cells?

We did not systematically compare anterior and posterior BLA cfos⁺ neurons or specifically focus on the anterior BLA. However, we observed learning-induced cfos⁺ neurons in both regions during recording, exhibiting synaptic potentiation (increased AMPA/NMDA ratio). This observation has been added to the Results section: “**GFP⁺ cells were observed throughout the anterior to posterior BLA**” (Page 2, line 84)

6) Figure 2–figure supplement 3F: The difference in spike numbers between conditions is not visually apparent. Please provide a better representative trace to support this data.

There is no difference in spike frequency, but the amplitudes of mEPSCs differ. “mEPSCs from CK2-D⁺ neurons have smaller amplitudes, but no difference in frequency, compared to Ck2-D⁻ neurons, suggesting post-synaptic depression.”

7) Figure 3–figure supplement 4C: The images of CS and CS+US conditions should represent similar levels along the anteroposterior axis of the BLA for better comparison.

We regret that we could not provide better examples. The current coordinates are approximately -1.5 mm (AP to bregma) for the CS group and -1.8 mm for the CS+US group, which we consider to be at similar levels.

8) Discussion A recent study (Wilmot et al., 2024) highlighted the limitations of the cFos-shEGFP transgenic mouse line (Jackson Laboratory, stock # 018306) due to cFos overexpression in the hippocampus. Could the authors discuss whether such overexpression might also occur in the BLA and address how this potential limitation could affect the interpretation of their data?

We appreciate the reviewer’s comment, and we are aware of the limitations recently described in Wilmot et al., 2024. However, we have no reason to believe that such limitations would impact any of the experiments described in the current manuscript. In Wilmot et al., 2024 the authors reported an apparent increase in the expression of the cfos protein when quantified using immunohistochemistry. They demonstrate that this apparent increase is due to the expression of the fusion protein C-Fos-Exon1-GFP. Exon 1 contains the N-terminus portion of the cfos protein and the authors suggest that the use of commercial antibodies directed at this portion of the cfos protein may bind to both the fusion protein and endogenous cfos, therefore giving the appearance that there is more cfos in the cFos-shEGFP mouse line. Importantly, the authors did not detect any evidence of altered expression for endogenous cfos (by quantifying endogenous Exon1 and Exon3), or other immediate early genes (Arc and Zif268), and did not find any behavioral abnormality in a contextual fear conditioning task in the cFos-shEGFP mouse line.

In our study, we did not perform immunohistochemistry in cFos-shEGFP mice but recorded from GFP⁺ neurons following behavior, observing specific synaptic changes (Figure 1A-E). We further validated this result using a different mouse line 7 days after CFC (Figure 1F-H), confirming that the observed effect was not an artifact of a specific transgenic mouse line. While we appreciate the reviewers' concerns regarding potential limitations of the cFos-shEGFP mouse line, we do not find it necessary to address them in this manuscript.